# Incommensurate grain-boundary atomic structure

Takehito Seki [1,2] ✉, Toshihiro Futazuka [1], Nobusato Morishige[3], Ryo Matsubara[4], Yuichi Ikuhara [1,5] & Naoya Shibata [1,5,6] ✉

Grain-boundary atomic structures of crystalline materials have long been believed to be commensurate with the crystal periodicity of the adjacent crystals. In the present study, we experimentally observed a Σ9 grain-boundary atomic structure of a bcc crystal (Fe-3%Si). It is found that the Σ9 grain-boundary structure is largely reconstructed and forms a dense packing of icosahedral clusters in its core. Combining with the detailed theoretical calculations, the Σ9 grain-boundary atomic structure is discovered to be incommensurate with the adjacent crystal structures. The present findings shed new light on the study of stable grain-boundary atomic structures in crystalline materials.

Silicon steel (Fe doped with a few % of Si) has been utilized in many industrial applications, including in the core materials of generators, transformers, and motors. To obtain high magnetic induction and low core-loss properties, grain-oriented silicon steel with strong {110} ⟨001⟩ texture (known as Goss texture) has been fabricated through complex thermo-mechanical processes since its seminal invention by Goss in 1934[1]. The high selectivity of Goss texture is generally attributed to the abnormal grain growth of Goss grains during secondary recrystallization. However, the key factors determining the abnormal grain growth are still a matter of conjecture. Some studies suggested that low-energy grain boundaries (GBs) with specific crystallographic orientation relationships to Goss grains (such as the Σ9 orientation in coincidence site lattice theory) may be the origin of the abnormal grain growth because low-energy GBs are considered to easily pass through secondary-phase pinning particles[2,3]. Some other studies proposed that high-energy GBs surrounding Goss grains may be crucial because their high diffusivity can coarsen or dissolve the pinning particles and result in high GB mobility[4]. However, such discussions critically lack experimental characterization of the actual GBs. Therefore, the atomic structure, energy, and migration mechanisms of GBs in silicon steel are not yet well understood.

Atomic-resolution scanning transmission electron microscopy (STEM) is a powerful tool for directly characterizing atomic structures and chemistry of GBs in metals and ceramics[5–7]. Recently, GB migration at the atomic scale has been experimentally observed in oxide materials[8,9], revealing successive transformations between different stable and metastable GB structures during its migration. However, direct observation of GB atomic structures in iron and steel by STEM is extremely challenging. This is because obtaining atomic-resolution images requires placing the samples in the high magnetic field of the magnetic objective lens (typically > 2 Tesla). As iron and steel are ferromagnetic, the strong lens magnetic field can considerably alter or destroy their magnetic and physical structures. In addition, the optical conditions needed for atomic-resolution imaging are disturbed by the magnetic field of the sample. Thus, experimental studies on GB atomic structures in iron and steel have been seriously hampered, and most atomic-scale GB studies are limited to theoretical approaches.

In recent years, a new objective lens system which realizes magnetic-field-free sample environment has been developed[10]. Using this objective lens system combined with a higher-order aberration corrector, atomic-resolution STEM imaging of silicon steel is becoming possible[10]. Moreover, using this new electron microscope, magnetic fields of Fe atoms in α-Fe$_2$O$_3$ crystal can be visualized in real space[11]. In the present study, the atomic structures of Σ9 GBs in silicon steel are directly observed using the magnetic-field-free atomic-resolution STEM. Moreover, the atomic structures of the same orientation GBs

[1]Institute of Engineering Innovation, School of Engineering, The University of Tokyo, Yayoi 2-11-16, Bunkyo-ku, Tokyo 113-8656, Japan. [2]PRESTO, Japan Science and Technology Agency, Kawaguchi, Saitama 332-0012, Japan. [3]Kyushu R&D Laboratory, Nippon Steel Corporation, 1-1 Tobihatacho, Tobata-ku, Kitakyushu-shi, Fukuoka 804-8501, Japan. [4]Steel Research Laboratories, Nippon Steel Corporation, 20-1 Shintomi, Futtsu-shi, Chiba 293-8511, Japan. [5]Nanostructures Research Laboratory, Japan Fine Ceramics Center, 2-4-1 Mutsuno, Atsuta-ku, Nagoya 456-8587, Japan. [6]Quantum-Phase Electronics Center (QPEC), The University of Tokyo, Hongo 7-3-1, Bunkyo-ku, Tokyo 113-8656, Japan. ✉e-mail: seki@sigma.t.u-tokyo.ac.jp; shibata@sigma.t.u-tokyo.ac.jp

are theoretically modeled using molecular dynamics (MD) simulations and density functional theory (DFT) calculations. It is found that the Σ9 GB atomic structure shows vast structural reconstruction and forms dense packing of icosahedral clusters in its core, forming an unusual GB atomic structure incommensurate with the bulk body-centered cubic (bcc) structure.

## Results
### STEM experiments
In this study, we prepared two Fe-3mass%Si bicrystals with Σ9 symmetric tilt GBs, whose interface planes were {221} and {114}, respectively. Details on the fabrication of bicrystals and preparation of TEM specimen are provided in the Methods section. The annular dark-field (ADF) STEM images of Σ9 {221} and {114} symmetric tilt GBs, observed along the [110] axis, are shown in Fig. 1 and Supplementary Fig. 1, respectively. Unique periodic atomic structures along the GB planes are clearly visualized. The Σ9 {221} GB structure observed herein differs completely from the ones predicted by previous theoretical calculations, whereas the observed Σ9 {114} GB structure is highly similar to those predicted by the previous theoretical calculations[12–19]. To confirm the accuracy of the previous theoretical studies on the Σ9 {221} GB, we independently performed DFT calculations based on the so-called γ-surface method, which is commonly adopted for exploring the most stable GB structure via rigid body translation of adjacent crystals. As shown in Fig. 2a, the stable structure obtained using the γ-surface method was the same as that predicted by the previous studies[12–19]. To evaluate the effect of Si atoms, we acquired the electron energy loss spectra (EELS) of the $L_{2,3}$ edge from the GB and the bulk regions (Supplementary Fig. 2). Si segregation was not clearly detected by EELS. As will be discussed later, the observed GB structure can be well reproduced as the most stable structure by theoretical calculations without considering Si dopants, i.e. in the pure bcc Fe case. Moreover, it is predicted that Si atoms do not segregate to the most stable Σ9 {221}GB structure by DFT calculation (Supplementary Fig. 3). Therefore, we conclude that the Si solute atoms do not have a strong effect on the formation of the observed Σ9 {221} GB atomic structure. It is suggested that the

previous theoretical studies failed to predict the observed structure because the γ-surface method may not sufficiently cover the possible candidates for the stable GB atomic structures. In recent years, it has been pointed out that the γ-surface method cannot comprehensively predict the most stable GB atomic structures even in elemental copper[20].

### Theoretical search for stable structure
To account for the numerous structural candidates for finding the most stable GB structure, we employed the simulated annealing technique in MD simulation using angular dependent potential (ADP)[21]. In simulated annealing, GB structures can be explored in a random manner[22,23]. We assumed various random initial structures with different numbers of atoms at the GB core region and systematically increased the simulation cell sizes along the [110] axis. All MD simulations were performed with gradually decreasing temperatures to find the most stable GB atomic structures. To avoid trapping to local minimum states, 50 independent simulated annealing calculations were performed to identify the most stable GB structure. Figure 2e shows a plot of the stable GB energies versus simulation cell sizes along the [110] axis. Although the stable GB structure predicted in the previous studies is uniquely obtained with the $1 \times 1$ cell size, more stable structures with different atomic configurations are obtained with $1 \times 3$, $1 \times 5$, and $1 \times 8$ cell sizes (the structure with $1 \times 6$ cell size is identical to that with $1 \times 3$). The most stable structure is that obtained with the $1 \times 8$ cell size with minor differences in energy. These structures are highly similar to each other and their projected atomic column positions along the [110] axis are in good agreement with the experimental ADF STEM image (Fig. 2d, Supplementary Fig. 4). In these structure models, light and dark blue circles represent the Fe atoms located in atomic layers of different depths along the [110] axis. As shown in Fig. 2b, the GB core structures (indicated by the gray line) can be described by the alternating stack of distorted pentagons (indicated by the dark and light blue lines) on the same atomic layers along the [110] axis, with one atomic column at their center (indicated by the red circles). Further relaxations in the DFT calculations did not show significant changes to the obtained structures (Supplementary Fig. 5). In addition, the simulated annealing with larger cell sizes of $2 \times 3$, $2 \times 5$, $2 \times 8$ and using recent Neural-Network-Potential (NNP)[24] were also performed. NNP has more flexible function form than ADP, and a well-tuned NNP using DFT results can achieve higher energy accuracy. Under the same potential models, the obtained building units and GB energies are almost identical to those of cell sizes $1 \times 3$, $1 \times 5$, and $1 \times 8$, respectively (Supplementary Fig. 6 and Table 1). In NNP, the absolute energy values become comparable to the DFT results, showing better energy accuracy. However, since the structure and energy tendency are the same, we use ADP results for the following analysis.

### Investigation on stable structures
To clarify the origin of the stability of the obtained GB structure, we investigated its three-dimensional atomic configurations. The orthogonal views of the stable GB core structures for the $1 \times 3$, $1 \times 5$, and $1 \times 8$ cell sizes are respectively shown in Fig. 3a–c. The alternatively stacked pentagons have approximately constant periodicity along the [110] axis, whereas the central atoms represented by the red circles have larger periodicities along the [110] axis. Based on their positions along the [110] axis, the central atoms can be classified into two types: those located at the interlayer positions (i) and those located at the center of the pentagons (p) as shown in Fig. 3a–c. The variety of the stable GB structures can be represented by the different sequences of the central atoms. The structures for $1 \times 3$, $1 \times 5$, and $1 \times 8$ cell sizes are represented by the sequences $[i_4 p]$, $[(i_3 p)_2]$, and $[(i_3 p)_2 (i_4 p)]$, respectively, where the subindices denote the number of repetitions. Here, we find three polyhedral building units[25] for the stable GB core structures: $Fe_{13}$ icosahedral cluster (I), double icosahedral cluster (D), and double

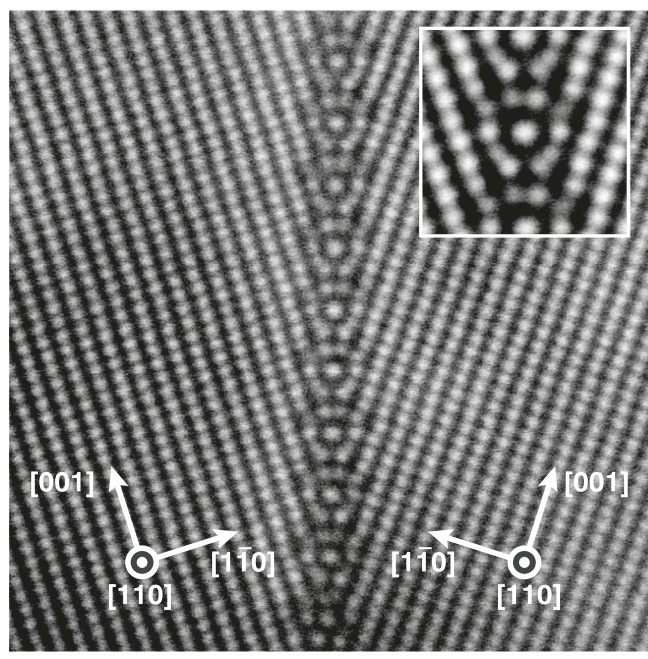

**Fig. 1 | ADF STEM image of Σ9 {221} symmetric tilt GB of the Fe-3mass%Si bicrystal.** The inset is the averaged image of the GB structure units. The scale bar represents 1 nm.

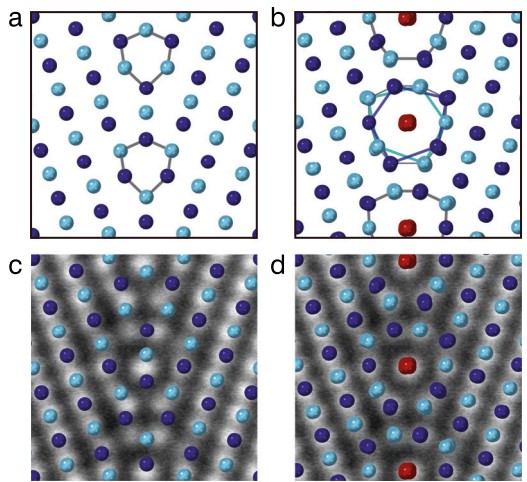
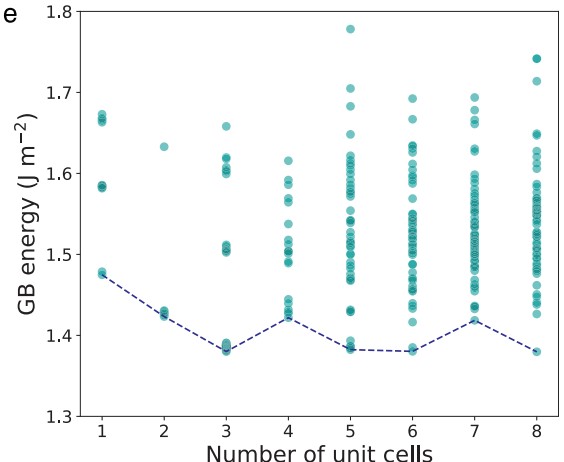

**Fig. 2 | Energetically stable Σ9 {221} GB structures predicted by theoretical calculations. a, b** The most stable GB structures derived from DFT calculations based on the γ-surface method and from MD calculations based on the simulated annealing method, respectively. **c, d** The experimental ADF STEM images of the GB structure units superimposed with the predicted structures shown in **a** and **b**, respectively. **e** Calculated GB energies versus simulation cell sizes along the [110] viewing direction using simulated annealing. The most stable GB energies in each simulation cell size are connected by the dashed line.

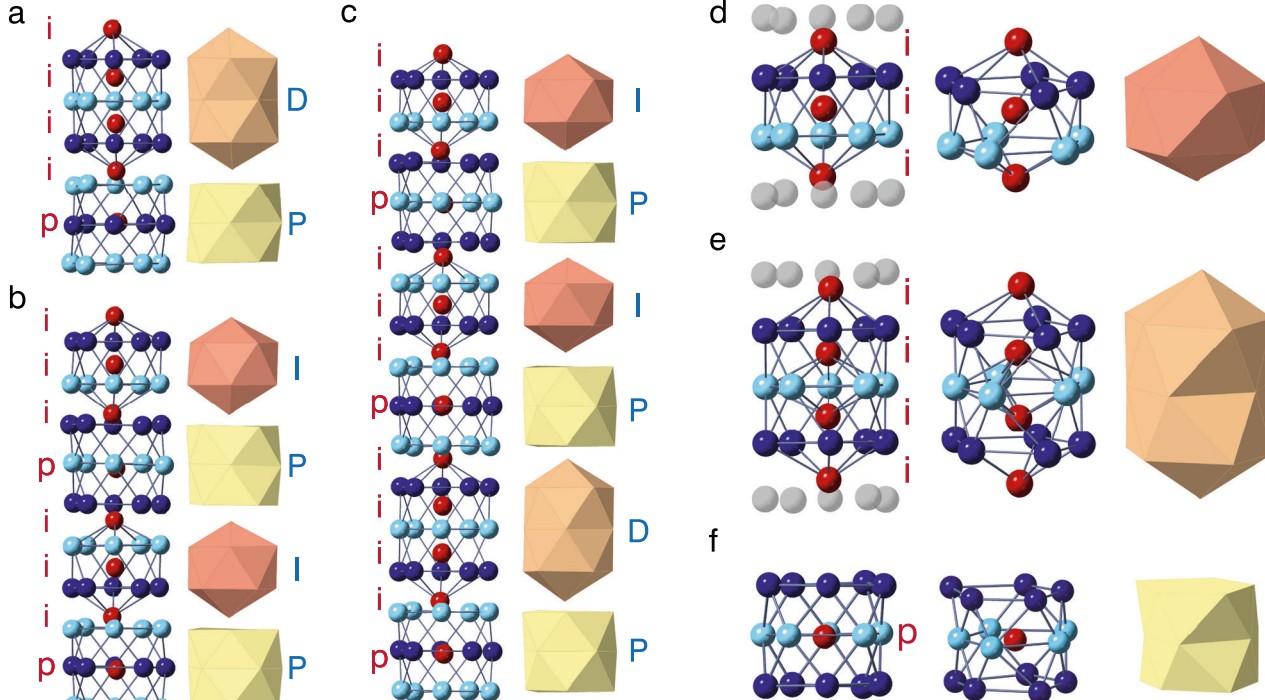

**Fig. 3 | Orthogonal views of the stable GB core structures described by their building units. a–c** Orthogonal views of the stable GB core structures derived by simulated annealing and their polyhedral model for the 1×3, 1×5, and 1×8 cell sizes, respectively. Building units of the stable GB core structures: **d** icosahedral cluster, **e** double icosahedral cluster, and **f** double pentagonal antiprism. The stable GB core structures for 1×2, 1×4, 1×7 cells are shown in Supplementary Fig. 11.

pentagonal antiprism (P), which are respectively shown in Fig. 3d–f. These building units correspond to the sequences of the central atoms of $i_3$, $i_4$, and p, respectively. Hence, the building unit sequences [DP], [IP], and [(IP)$_2$DP] compose the most stable structures of 1×3, 1×5, and 1×8 cell sizes, which are shown in Fig. 3a–c. To check the periodicity of the building units, we performed MD simulation with 1×32 cell size (Supplementary Fig. 7). It is confirmed that the GB structure is not the repetition of shorter-period structures. Thus, the Σ9 {221} GB forms long-period atomic structure along the [110] axis. This structural characteristic is highly distinct from the short-period structure units

commonly found in many metal and ceramic GBs[26]. It is also noted that such long-period structure is very different from the common heterophase boundaries with periodically fixed misfit dislocations and coherent regions in between.

The icosahedral cluster is well known as a kind of dense packing and universal stable structure formed in many metals: e.g., Frank–Kasper phases[27,28] quasicrystals[29], and metallic glasses[30]. However, the icosahedral clusters hardly form in simple bulk metals because the icosahedral symmetry is incompatible with crystal periodicity. In the present case, the constraint of structural periodicity of the bulk crystal

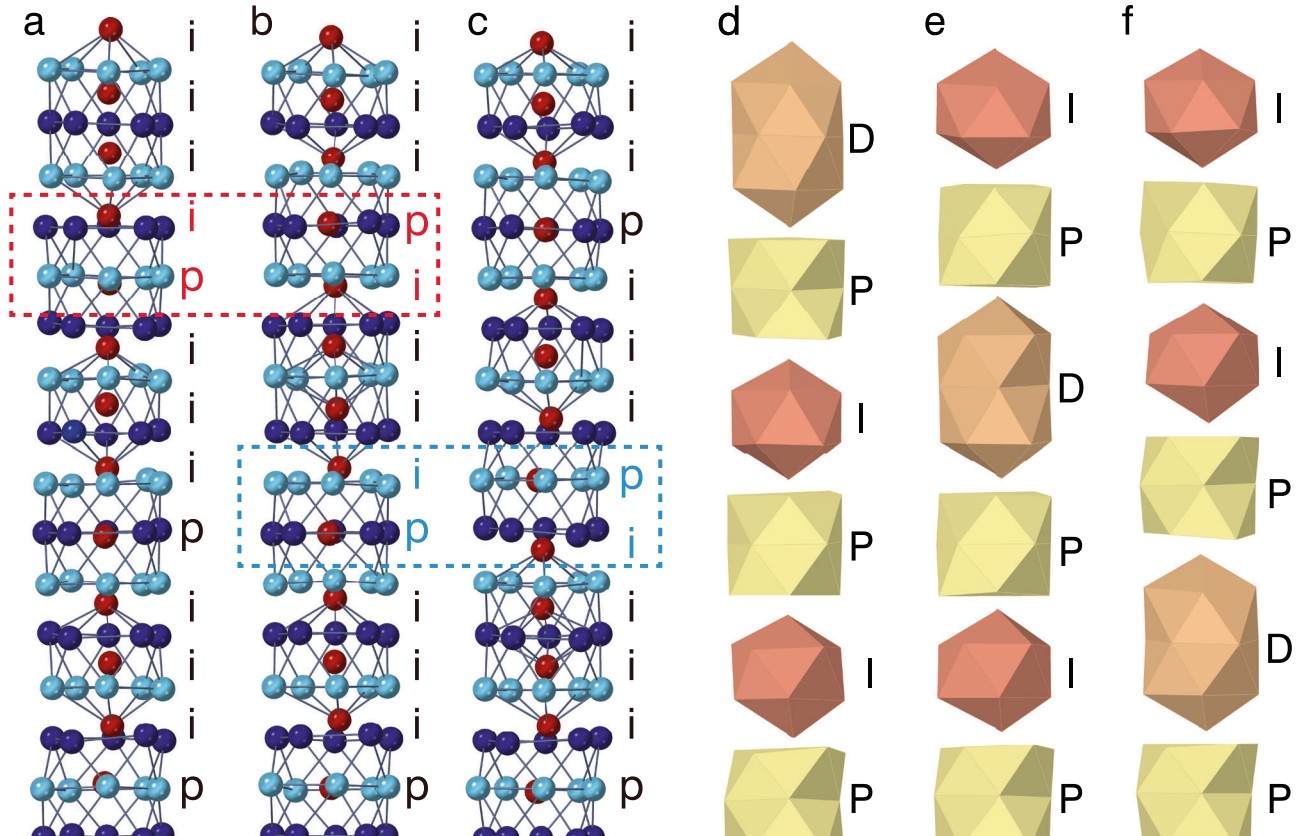

**Fig. 4 | Snapshots of the GB core structures simulated by MD with the 1 × 8 cell under a constant temperature of 300 K. a–c** Orthogonal views of the snapshots. The position category of the center atoms: 'i' or 'p' (see text), is shown beside the structure models. The elapsed times corresponding to **b** and **c** are 500 and 1000 fsec with **a** as the reference. **d–f** Polyhedral models of the snapshots. The name of the building units: 'I', 'D', or 'P' (see text), are shown beside the polyhedral models.

structure is locally absent at the GB. Therefore, the formation of icosahedral clusters with long-period structure is possible in the Σ9 {221} GB. If the icosahedral clusters were to form with the same periodicity as the surrounding bulk environment, the icosahedrons must be compressed along the [110] direction as shown in Supplementary Fig. 8. This structure is metastable because the distance between the center atoms along the [110] axis becomes too close. In the long-period structures with larger simulation cells, clusters with more regular icosahedral shape can be formed and parts of the center atoms are located at the center of the pentameric ring. These structures appear to maximize the number of stable icosahedral clusters along the GB core.

If the average distance between the central atoms is assumed to be the same as the stable Fe-Fe distance in the bulk, the relative density of the central atomic column with respect to that of the atomic columns in the bulk along the [110] direction is expected to be $2\sqrt{6}/3 \simeq 1.63$ (Supplementary Fig. 9). This value is very close to the most stable relative density in the 1 × 8 cell: $13/8 = 1.625$. To approximate the irrational relative density of $2\sqrt{6}/3$, the long-period structure is essential. This may explain the stability of the long-period structure in the present system. This situation is simplified by the Frenkel−Kontorova model, wherein a linear chain of atoms interacts with an external periodic potential[31]. In this model, the linear chain of atoms is displacively modulated by the external potential, and the position of the $n$th atom may be written as[32]

$$x_n = na + \phi + f(na + \phi), \qquad (1)$$

where $a$ is the average interatomic distance, $\phi$ is a phase, and $f$ is a periodic and continuous modulation function with the same period as the external potential. If the periodicity of the external potential is irrational to $a$, the modulation is incommensurate and does not have a simple periodicity. Because the modulated structures are equivalent for all phase $\phi$, the atomic chain is not locked to the external potential[33]. In relation to the phase, an additional lattice excitation−the so-called phason−arises in the incommensurate structure. The phason can induce structural fluctuations and contribute to the free-energetic stability of incommensurate phases due to entropy gain. This unique entropic stabilization mechanism has been extensively studied in incommensurate phases and quasicrystals[34]. Incommensurate structures generally undergo a phase transition to commensurate structures at low temperatures[33]. As for quasicrystals, specific heat exceeds the Dulong−Petit value at high temperatures due to the phason[35]. These experimental results indicate that the entropy gain by the phason is comparable to the phonon entropy and contributes significantly to the phase stability in incommensurate and quasiperiodic structures.

### Theoretical investigation on structural fluctuations

We performed MD simulations for the stable GB structure of the 1 × 8 cell at a constant temperature and explored the possibility of dynamic structural fluctuations. Supplementary Video 1 and Fig. 4 show the movie of dynamic structural fluctuations and their snapshots, respectively. The times corresponding to the second and third snapshots−500 and 1000 fs−are with reference to the first snapshot. Although the local sequence along the [110] direction is exchanged as 'ip' to 'pi' as indicated by the red and blue characters, all the structures have the equivalent stable sequence of the central atoms $[(i_3p)_2(i_4p)]$, i.e., the sequence of the building units $[(IP)_2DP]$, with different phases of the modulation as shown in Fig. 4d−f. Thus, the MD simulation shows that the stable GB structure in the 1 × 8 cell fluctuates back and forth between equivalent structures at finite temperatures. The similar MD

simulation with the larger cell size of $1 \times 32$ shows that the overall GB atomic structure can be different before and after the fluctuation in the larger cell sizes (Supplementary Fig. 10). As an analogy to the previous studies of incommensurate phases and quasicrystals, this behaviour may contribute to the stabilisation of the GB atomic structure at finite temperature through entropy gain. However, further detailed theoretical analysis should be needed to identify the energy stabilisation effects by the dynamic structural fluctuation, which will be our future study.

## Discussion

In summary, we performed direct atom-resolved imaging of Σ9 GBs in silicon steel using magnetic-field-free atomic-resolution STEM. The atomic structure of Σ9 {221} GB is found to be completely distinct from the structures predicted by numerous previous theoretical calculations. Based on the observed structure, we performed extensive theoretical calculations and found that the GB atomic structure shows incommensurate nature with respect to the bulk bcc structure. The formation of this unexpected structure should give a new insight into the study of stable GB atomic structures in crystalline materials.

## Methods

### Bicrystal fabrication

Electrolytic iron and pure silicon were melted in vacuum by high-frequency induction heating with the composition of Fe-3mass%Si. The prepared ingot was annealed at 1350 °C for 48 h to grow coarse grains with a size of several tens of millimeters. The crystal orientations of the coarse grains were measured by the back-reflection Laue method and single crystals were then cut from the ingot. The bonding surfaces were mechanically polished, and two pairs of single crystals were diffusion-bonded at 1200 °C for 24 h in 100% hydrogen atmosphere to form Σ9 {221} and Σ9 {114} bicrystals.

### TEM specimen preparation

The fabricated bicrystals were cut into a rectangular shape with a size of several millimeters and then mechanically polished to a thickness of approximately 100 μm. The polished specimens were further thinned (with cooling the specimens) by Ar ion beam using a cryo ion slicer (JEOL IB-09060CIS). Prior to the STEM observation, the TEM specimens were further polished (with cooling the specimens) by Ar ion beam to remove the surface oxide layer using an ion polishing machine (Fischione, Model 1051). The polished specimens were transferred to the TEM without exposure to air as follows: First, the specimens were transferred from the ion polishing machine to a glovebox using a vacuum transfer capsule. Second, the specimens were loaded in a vacuum/inert-gas transfer TEM holder (JEOL Ltd.) in the glovebox filled with argon gas. Third, the argon gas in the TEM holder was evacuated to below $1 \times 10^{-4}$ Pa and the specimens were then heated at approximately 80 °C for 12 h to remove the contaminations using a high-vacuum TEM holder station (Mel-build LF4-CUBE). Finally, the specimens were transferred to the specimen chamber in the TEM.

### STEM observations and image simulation

The STEM observations were conducted using an electron microscope (MARS: JEM-ARM200F) equipped with a magnetic-field-free objective lens, operated at 200 kV. The probe forming aperture semiangle and the inner collection semiangle of the ADF detector were set to 25 and 50 mrad, respectively. To suppress the image distortion due to the specimen drift, ten ADF images were sequentially acquired and then averaged after image registration based on the cross-correlation.

The STEM image simulation was performed based on the multislice method with the frozen phonon model. The microscope parameters were set to be consistent with the experimental parameters. The thickness of the specimen was assumed to be 10 nm. The effects of the effective source size and the chromatic aberration were considered by convoluting the Gaussian function with a full width at half maximum of 0.7 Å.

To investigate Si segregation at the GB, EEL spectra were acquired from the bulk and GB regions. The spectra were acquired by scanning the electron probe in the regions at a distance of approximately 3 nm away from the grain boundary and within approximately 1 nm in regions on the grain boundary core.

### Theoretical calculations

In the γ-surface method, the initial structural models were prepared from two monocrystalline slabs with the {221} free surfaces with the orientation relationship of Σ9. To form the GB, the distance between the surfaces of two slabs was set to 1.0 Å and one slab was translated along the in-plane direction of the surface with 0.2 Å step. A total of 589 (=19 × 31) initial structural models were relaxed by MD calculations with angular-dependent potential[21] in the simulation package LAMMPS[36] in the isothermal-isobaric ensemble at 0 K and 0 atm. The most stable structure was further relaxed via first-principles calculation based on the DFT. The calculation was based on the projector augmented wave method implemented in VASP[37,38] with the GGA-PBE functional[39]. The spin polarization was considered in the calculation with the initial magnetic moment of 3.0 $\mu_B$ for each Fe atom. The relaxed structure is shown in Fig. 2a.

The simulated annealings[22,23] were performed based on MD calculations. In the present calculation, we used the simulation cell with double-grain boundaries instead of slab model. In this model, there are two equivalent GBs in the supercell, and no vacuum layers are introduced. Then, we constructed the N-fold superstructures (N = 1,2,…,8) by multiplying the primitive Σ9 {221} GB cell along the rotation axis. We selected the GB separation distance in the supercell to be about 16.90 Å because GB energies show almost no change even if we increase the separation distance (Supplementary Fig. 12). To investigate GB structures with various atomic densities, Fe atoms were removed from the constructed superstructures up to N−1 atoms for each N-fold superstructure. We considered a total of 32 superstructures with various cell sizes and atomic densities. Furthermore, 50 initial configurations were generated for each superstructure by heating the core region of GB (6.0 Å width) at 20,000 K and extracting the structures every 0.5 ps. Next, all the atoms in the initial structures were respectively annealed with a gradual reduction in temperature from 700 K to 0.1 K in the NPT ensemble using the Nose-Hoover thermostat[40,41]. In the temperature range 700 K to 300 K, the cooling rate was set to −10 K/ps and in the range 300 K to 0.1 K, the cooling rate was set to −100 K/ps. The energy values of all final structures are plotted in Fig. 2e. To confirm the validity of the obtained structures based on the simulated annealing, we further performed first-principles calculations under conditions identical to those used in the γ-surface method. The stable structures obtained through the simulated annealing for 1 × 1, 1 × 3, 1 × 5, and 1 × 8 cells were further relaxed by DFT, as shown in Supplementary Fig. 5. Since the initial magnetic moments and configurations should strongly influence the final magnetic structure and the GB energy, our obtained GB energies by DFT shown in Supplementary Fig. 6 and table 1 can be trapped to the metastable ones. To demonstrate this possibility, we performed additional DFT calculations as shown in Supplementary Fig. 13. It is seen that the GB energies after relaxation strongly depend on the initial magnetic moment and configuration in the present DFT. Because of the limited computational resources, finding the most stable magnetic structures by DFT for the present GB is beyond the scope of the present study.

To investigate the possibility of Si segregation to the Σ9 {221} GB, we evaluated the segregation energy of Fe-substitutional Si using DFT calculation. The segregation energy of $i$ site at the GB, $\Delta E_{seg}[i]$, is defined as

$$\Delta E_{seg}[i] = \Delta E_{def}^{GB}[i] - \Delta E_{def}^{bulk}, \qquad (2)$$

where $\Delta E_{\mathrm{def}}^{\mathrm{GB}}[i]$ and $\Delta E_{\mathrm{def}}^{\mathrm{bulk}}$ are the defect formation energies of $i$ site in the $\Sigma 9\ \{221\}\ (1 \times 3)$ GB and in the bulk, respectively. To remove the elastic interaction between the periodic images of Si, the distance between Si atoms is set to 12.29 Å and 12.09 Å for the bulk and GB supercell, respectively. Supplementary Fig. 3a shows the structure of $\Sigma 9\ \{221\}$ GB, and we evaluated the segregation energies of all atomic sites in the core region (labelled A to F). Supplementary Fig. 3b shows the segregation energies of Si as a function of distance from the center of the GB. The minimum segregation energy of Si is 0.84 eV at the site B (shown with red arrow). However, the segregation energies of all the sites are positive, indicating that the Si atoms are energetically less stable in these sites than in bulk sites. Thus, the present calculation suggests that Si would not segregate to the $\Sigma 9\ \{221\}$ GB.

## Data availability

The presented data are available at GitHub[42].

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

## Acknowledgements

We thank S. Sugiyama for useful discussions and preliminary theoretical calculations. We thank K. Murakami and M. Takahashi for useful discussions. This work was supported by JST ERATO (grant number JPMJER2202) and the JSPS KAKENHI (grant numbers JP20H05659 and JP19H05788). A part of this work was supported by the JST SENTAN (grant number JPMJSN14A1), the JSPS KAKENHI (grant number JP17H06094), and "Advanced Research Infrastructure for Materials and Nanotechnology in Japan (ARIM)" of the Ministry of Education, Culture, Sports, Science and Technology (MEXT) (grant number JPMXP1222UT0289). T.S. acknowledges support from JST-PRESTO (grant number JPMJPR21AA), JSPS KAKENHI (grant number JP20K15014), and the Kazato Research Foundation. T.F. acknowledges the support from Grant-in-Aid for JSPS research fellow (grant number JP22J15213).

## Author contributions

T.S. and N.S. designed the study and wrote the paper. T.S. performed the STEM experiments and image simulations. T.F. performed the theoretical calculations. N.M. and R.M. fabricated the bicrystal specimens. Y.I. contributed to the discussions and suggestions. T.S. and N.S. directed the entire study.

## Competing interests

The authors declare no competing interests.
