## [Peer Review File · Nature Communications]

Incommensurate grain-boundary atomic structureREVIEWER COMMENTS

Reviewer #1 (Remarks to the Author):

The paper "Incommensurate GB atomic structure" by Seki et al reports interesting experimental results on a Sigma 9 GB in Fe(Si) by atomic resolution TEM. The structure found experimentally is different to previous atomistic simulations and the authors showed that the cell size matters in the atomistic GB structure simulations. Up to this point, the story line of the paper is consistent and strongly underpinned by experimental and theoretical results. However, the additional claims regarding (i) the phason, (ii) the entropic stabilization, and (iii) the origin of abnormal grain growth are only speculations and not well related to experimental and computational findings of the paper.

@ (i): The authors claim that an irrational relationship between crystal periodicity and GB periodicity leads to a "decoupling" of the two symmetries and thus the Frenkel–Kontorova model should apply. But there is nothing proven with data. In reality, I believe that reconstructions or line defects would re-establish a periodicity like at heterophase boundaries where mismatch is also leading to an irrational aspect ratio of lattice plane distances, but misfit dislocations would be incorporated so that coherent regions (partly strained) are established. This point would require discussion and maybe even cross-sectional STEM studies to observe the lattice planes across the GB.

Furthermore, from an atomistic simulation point of view the authors tried a handful of GB reconstructions; maybe the periodic one is just missing in your data set (maybe it is 1x2, 1x4, 1x7, 2x2, ...).

Finally, can you also exclude artefacts due to fast cooling in your simulations?

@ (ii): I didn't see a calculated or measured entropy – so it is hard for me to judge on the entropic stabilization (or I didn't understand the point sufficiently well enough the authors wanted to make here).

@ (iii): To make the claim that this GB structure is the origin for abnormal grain growth would need to be supported by grain growth experiments and/or GB migration simulations. According to your EELS data, I could also claim that the GB is not pinned as no Si is enriched at the GB and thus speculate that it is more mobile than other GBs.

Reviewer #2 (Remarks to the Author):

1. This is an interesting and stimulating paper in which state-of-the-art experimental high-resolution microscopy of two different σ_9 grain boundaries in bcc Fe are compared. The presence of 2% Si in the experimental samples is ignored for the purposes of analysis with theoretical models, which is a reasonable for a first study of this kind, which suggests that the free energy of phasons is an important stabilizing mechanism in the $\Sigma_9\{221\}$ grain boundary.
2. The upper-case notation in Fig.3 used for labelling sub-figures, is inconsistent with the lower-case notation in the caption. The figure would be less confusing if sub-figures were also labelled in lower case. Upper case letters I, D, P are also used in Fig.3 to label structures within sub-figures A, D, C, but not in sub-figures D, E, F. This should be made consistent within the figure and also consistent with the notation in the body of the text. The same inconsistent use of upper or lower case occurs in the Extended data figures, but it is less confusing there.
3. A particular problem with the paper is that there is insufficient consideration of the precision of the calculations, meaning the sources of error in grain boundary energies (and structures) apart from the well-known inaccuracy of empirical potentials or DFT. We are told that this is a slab calculation with periodic boundary conditions parallel to the grain boundary and free surfaces on the outside of the slab. Since the boundaries are symmetric the two free surfaces are the same. Presumably the atoms at the free surface and for some depth into the slab are not relaxed, or perhaps they are. In either case I can't find any discussion of the influence on the results of the thickness of the slab, and from experience I know that this should be much greater than the periodicity along the boundary. Is the system also periodically repeated perpendicularly to the grain boundary, and, if so, how thick is the vacuum region? The paper includes an investigation of varying the supercell size along the [110] axis but not along an orthogonal direction parallel to the boundary, which should be no less worthy of testing.
4. With reference to the relaxed structures obtained with the 1x3, 1x5 and 1x8 repeat units after simulated annealing we read (line 117): "Further relaxations in the DFT calculations did not show significant changes to the obtained structures". This is curious, given that, according to Extended Data Table 1, although the energy of the 1 x 3 is lower than that of the 1 x 5 by only 5 mJ/m² when both are calculated with simulated annealing, the further relaxed DFT energy of the 1 x 3 is 101 mJ/m² lower than that of the 1 x 5. I believe 5 mJ/m² would be within the error bars of the calculation, and structures that appear to be separated by this energy could well be equivalent. However, 101mJ/m² stands out as indicating a significant difference in structure. For example, stacking fault energies are typically of this order of magnitude. This is a particular example for which more testing and reporting of the precision of calculations might be illuminating.
5. With reference to the phason free energy, there is no doubt that the phasons contribute *something* to the stabilisation of the structure at finite temperature, but they make no contribution to its stabilisation in the zero temperature DFT calculations presented here. Even at zero temperature, phonon zero-point energies, not considered here, with or without phasons present, make a difference to the free energy.
6. No evidence is given to support the statement "Thus, the free energy of the $\Sigma_9\{221\}$ GB can be much lower than other GBs at finite temperatures"; that is to say, there is no data presented to indicate that "much lower" makes the difference between a metastable structure and a most stable structure. The role of phasons needs to be more quantitatively identified in the model grain boundary energy, even though in real

materials this would be experimentally very difficult. Note that anharmonicity and electronic excitations always cause the specific heat of a crystalline material to exceed the Dulong-Petit limit at high temperature, and quantisation of phonon energy cause it to be lower than this limit at low temperature, with the outcome that the Dulong-Petit limit is mostly observed as little more than a point of inflection in the specific heat versus temperature of a perfect crystal. And the free energy of perfectly periodic grain boundaries generally tends to decrease with temperature without phasons. So the statement on line 178: "As for quasicrystals, specific heat exceeds the Dulong-Petit value at high temperatures due to the phason" is not helpful to the case, without some quantitative information and comparison with the other effects that are relevant at finite temperature.

7. A minor typo in lines 115 and 125: alternative -> alternating

Reviewer #3 (Remarks to the Author):

The authors reported on an incommensurate grain-boundary atomic structure of $\Sigma_{9/221}$ STGB in Fe-Si alloy based on STEM and atomistic simulations. They also proposed an unconventional entropic stabilization mechanism at finite temperature due to an additional lattice excitation (i.e. phason). However, I disagree with most of their arguments. 1. the authors said there is no Si segregation in the GB (based on EELS). Thus the reported structure should be independent on Si segregation. This is hardly believed. 2. since they don't consider Si role in GBs, all simulations were conducted for PURE Fe. The authors can examine Si solubility in GB compared with in grains. If Si is preferred to segregated into GBs, then the author must prove Si segregation does not influence the B structure. 3. Since there is no reliable empirical potential for Al-Si system, all simulations should be done by using DFT. The authors should consider the size effect on DFT calculations, not only the unit cells, but also the size of the two grains because of either two free surfaces or double GBs in the model.

Reviewer #4 (Remarks to the Author):

I like your expertized work with target-oriented preparation of bicrystals, cutting edge STEM and detailed theoretical calculations to interpret the observations and consequences of them.

Your observation that proper understanding of the symmetric $\Sigma_{9/221}$ in bcc Fe, meaning finding its stable lowest energy state, needs multiple lattice periodic units along the tilt axis is remarkable. This justifies publication as it clearly demonstrates that the usual practice of simulating GB structures in rather small depth periodic units can be misleading for unfortunate choice of grain boundary. On the other hand, it might not be too surprising, considering the fact that all known investigations naturally use several lattice units perpendicular to the axis when constructing a supercell. To the bottom end, a GB is a two dimensional surface, for which it is likely that the supercell has to be extended in both lateral dimension to reasonably approximate the structure.

It is a very clear result that your newly identified structure, described as a sequence of

ikosahedral, double ikoshedral clusters and biprisyms, has a lower energy, convincingly matches to your STEM images, and your quantitative values on the GB excess energy appear reasonable. I can well understand that the equilibrium structure could be only found with the help of simulated annealing, while conventional DFT search missed it. But after it has been found, I do not understand that the corresponding DFT energies (third column of table in extended material) do not show the energy reduction with reference to 1x1. If there is no simple error in the presented numbers, the origin of this calculation must be better explained in the article to clarify the situation.

By contrast, I am less confident that the shown evidence for the phasons and the consequent stabilization of Gibbs energy by additional entropic contributions, is sufficient to be convincing. Yes, it is clear that the low energy configuration of the GB needs a longer sequence of different building blocks. Also the total length of this sequence might be incommensurate to the initial lattice periodicity, although I miss a real proof for this in the article. But most important, the three snapshots in Fig. 4 just essentially show three times the same structure, if one takes into account the periodic boundary conditions between top and bottom of the simulated volume. So what you interpret as a phason could simply be a numerical effect giving a slight shift to the position with respect to the position of the periodic boundaries.

If you would like to keep your statement of entropic stabilization and phasons, I feel that a controlled variation of the height of the simulated volume over a larger range, and quantitative comparison of the outcome with the Equation (referenced from 30) is necessary. Also a clear estimation/calculation of the amount of the entropic effect to the Gibbs energy would be required. As far as I understood, the presently applied theoretical methods are only suitable to calculate the energy/enthalpy. For calculation of the entropy or the Gibbs energy, either quasiharmonic approximation, switching hamiltonian methods (see e.g. doi:10.1016/j.actamat.2018.01.006) or similar should be applied to derive a clear prediction of the excess entropy of the GB.

Response to the Reviewer #1

We thank the reviewer for reviewing our manuscript. We are very gratified that the reviewer basically agrees with us on the significance of our findings. However, the reviewer raised several concerns and questions on our manuscript. We have carefully considered all the criticisms and comments raised by the referee, and revised our manuscript accordingly by incorporating new data in order to address all these concerns. The summary of our responses to the points raised by the reviewer and the corresponding changes made to the manuscript are italicised and interspersed between the reviewer's report below. In the revised manuscript, we highlight how we revised our manuscript by red colour font.

The paper "Incommensurate GB atomic structure" by Seki et al reports interesting experimental results on a Sigma 9 GB in Fe(Si) by atomic resolution TEM. The structure found experimentally is different to previous atomistic simulations and the authors showed that the cell size matters in the atomistic GB structure simulations. Up to this point, the story line of the paper is consistent and strongly underpinned by experimental and theoretical results. However, the additional claims regarding (i) the phason, (ii) the entropic stabilization, and (iii) the origin of abnormal grain growth are only speculations and not well related to experimental and computational findings of the paper.

We are gratified that the reviewer agrees with us on the significance of our paper. As will be shown later in detail, we have performed substantial computational analysis to further reinforce the points (i) and (ii). On the point (iii), the reviewer is correct that we are still in speculative discussion on the abnormal grain growth behaviors. Since understanding the actual grain boundary migration process is beyond our scope of this study, we have modified our discussion in the revised manuscript.

@ (i): The authors claim that an irrational relationship between crystal periodicity and GB periodicity leads to a "decoupling" of the two symmetries and thus the Frenkel–Kontorova model should apply. But there is nothing proven with data. In reality, I believe that reconstructions or line defects would re-establish a periodicity like at heterophase boundaries where mismatch is also leading to an irrational aspect ratio of lattice plane distances, but misfit dislocations would be incorporated so that coherent regions (partly strained) are established. This point would require discussion and maybe even cross-sectional STEM studies to observe the lattice planes across the GB.

It is well known that misfit dislocations are introduced in the heterophase boundaries to accommodate the different periodicity of the adjacent heterophase crystals. Misfit dislocations are introduced periodically and their positions are fixed and not mobile. The present GB structure is very different from such heterophase boundary cases. In the present case, the core Fe atom chain (shown as red) along the [110] direction is not commensurate with the surrounding bcc crystal periodicity along the [110] direction. From the MD simulation, only the core Fe atoms can be fluctuated along the [110] direction, while the surrounding bcc structure is kept fixed at finite temperatures. We

found that the GB energies are kept constant by the Fe atom fluctuation. This means that, at finite temperatures, the core Fe atom is always fluctuated along the $[110]$ direction by thermal energy and such fluctuated structure (not rigid one) becomes the stable structure. This atom fluctuation (loose structure) is the characteristic of the present GB and very different from the common heterophase boundaries with periodic misfit dislocations. The following sentence is now added to the revised manuscript as follows.

It is also noted that such long-period structure is very different from the common heterophase boundaries with periodically fixed misfit dislocations and coherent regions in between.

Furthermore, from an atomistic simulation point of view the authors tried a handful of GB reconstructions; maybe the periodic one is just missing in your data set (maybe it is 1×2 , 1×4 , 1×7 , 2×2 , ...).

The energetically most stable structures found in 1×2 , 1×4 , 1×7 cells are shown in Fig.R1. It is seen that the atomic structures of these GBs are basically similar to those in 1×3 , 1×5 , 1×6 , 1×8 cells in terms of the stacking of unit clusters. However, due to the cell size constraint, the connection between clusters need to introduce a new cluster, which we call pentagonal dipyramid cluster (Dp). Due to the introduction of Dp cluster, the GB energies of 1×2 , 1×4 , 1×7 cells become higher than those of 1×3 , 1×5 , 1×6 , 1×8 cells. Thus, if we increase the cell size, the non-periodic GB structure becomes energetically favorable than the short period ones, which is the characteristic feature of this $\Sigma 9$ GB.

Fig.R1 is included as Supplementary information as Fig.S10.

Fig.R1: The stable GB atomic structures of 1×2 , 1×4 , and 1×7 cells. **a**, **b** show the atomic structures of 1×2 (1×4) and 1×7 GBs observed along the $[110]$ axis. Note that the 1×2 and 1×4 GBs have the same atomic structure. The GB atomic structures viewed along the $[110]$ axis are similar to those of 1×3 , 1×5 , 1×6 , 1×8 cells. **c**, **d**, **e** show the atomic structures of 1×2 , 1×4 , and 1×7 GBs observed along the orthogonal direction. In these GBs, in addition to the icosahedral cluster (I) and the double-pentagonal antiprism (P),

the pentagonal dipyrmaid clusters (D_p) are formed in the GB core.

Finally, can you also exclude artefacts due to fast cooling in your simulations?

As the referee pointed out, the system can be trapped in a local minimum state due to the fast cooling. To avoid such trapping, we have made 50 independent simulated annealing calculations and identify the most stable GB structure. We comment on this point in the revised manuscript.

To avoid trapping to local minimum states, 50 independent simulated annealing calculations were performed to identify the most stable GB structure.

@ (ii): I didn't see a calculated or measured entropy – so it is hard for me to judge on the entropic stabilization (or I didn't understand the point sufficiently well enough the authors wanted to make here).

We evaluated the GB free energies, $F_{GB}(T)$, using the non-equilibrium Frenkel-Ladd method (D. Frenkel et al., J. Chem. Phys. 81, 3188 (1984), M. Watanabe et al., Phys. Rev. Lett. 65, 3301 (1990), R. Freitas et al., Comput. Mater. Sci. 112, 333 (2016)), which is a kind of switching Hamiltonian method. We performed all the free energy calculations with ADP potential. Fig.R2 shows the GB free energies of $\Sigma 9 \{221\} (1 \times 8)$, $\Sigma 9 \{221\} (1 \times 1)$, and $\Sigma 9\{114\}$ GBs as a function of temperature. The $\Sigma 9 \{221\} (1 \times 8)$ has much lower GB free energy in all the temperature range. The temperature dependences of the GB free energies are approximated with the straight lines. The GB entropies obtained from the slope of $F_{GB}(T)$ for $\Sigma 9 \{221\} (1 \times 8)$, $\Sigma 9 \{221\} (1 \times 1)$, and $\Sigma 9\{114\}$ are 2.37×10^{-1} , 1.39×10^{-1} , and $8.45 \times 10^{-2} \text{ mJ m}^{-2} \text{ K}^{-1}$, respectively. The $\Sigma 9 \{221\} (1 \times 8)$ GB has a much higher GB entropy than other GBs. The present results clearly show that the incommensurate GB structure has entropic stabilization effect at high temperatures. We include the Fig.R2 as the new Fig.4 and also included corresponding discussion in the main manuscript as follows.

To investigate the entropic contribution from the phason in the present GB, we evaluated the GB free energies, $F_{GB}(T)$, using the non-equilibrium Frenkel-Ladd method, which is a kind of switching Hamiltonian method. We performed all the free energy calculations with ADP potential. Fig. 4 shows the GB free energies of $\Sigma 9 \{221\} (1 \times 8 \text{ cell})$, $\Sigma 9 \{221\} (1 \times 1 \text{ cell})$, and $\Sigma 9\{114\}$ GBs as a function of temperature. The $\Sigma 9 \{221\} (1 \times 8 \text{ cell})$ has much lower GB free energy in all the temperature range. The temperature dependences of the GB free energies are approximated with the straight lines. The GB entropies obtained from the slope of $F_{GB}(T)$ for $\Sigma 9 \{221\} (1 \times 8 \text{ cell})$, $\Sigma 9 \{221\} (1 \times 1 \text{ cell})$, and $\Sigma 9\{114\}$ are 2.37×10^{-1} , 1.39×10^{-1} , and $8.45 \times 10^{-2} \text{ mJ m}^{-2} \text{ K}^{-1}$, respectively. The $\Sigma 9 \{221\} (1 \times 8 \text{ cell})$ GB has a much higher GB entropy than the other GBs. The present results clearly show that the incommensurate GB structure has entropic stabilization effect at finite temperatures.

Fig.R2: GB free energies as a function of temperature. The error bars correspond to the standard deviations of 10 independent simulations.

@ (iii): To make the claim that this GB structure is the origin for abnormal grain growth would need to be supported by grain growth experiments and/or GB migration simulations. According to your EELS data, I could also claim that the GB is not pinned as no Si is enriched at the GB and thus speculate that it is more mobile than other GBs.

We thank the reviewer for this comment. We have performed DFT calculation of Si segregation energies in this GB. The results are shown in Fig.R3. Here, we evaluated the Si segregation energies for all the Fe atomic sites in the core region (labeled A to F). Our results show that the segregation energies of Si become positive for all the GB sites, suggesting that Si atoms do not segregate to the present GB. These results are in agreement with our EELS experiment. We included the Fig.R3 as the Supplementary Fig.3. As the reviewer pointed out, these tendencies may also contribute to the GB migration process. However, since understanding the actual GB migration process is beyond our scope of this study, we have modified our discussion on the relation to abnormal grain growth in the revised manuscript as follows.

This unconventional structure and resulting energy stabilization mechanism may play a role in the microstructure evolution in the grain-oriented silicon steel sheets. In general, the pinning force from the secondary-phase particles is considered to be proportional to GB energies. Therefore, if the $\Sigma 9$ GBs have very low GB energies, they may retain high mobility even in the presence of secondary-phase particles. In this study, we experimentally and theoretically found a more stable atomic structure of the $\Sigma 9 \{221\}$ GB than the previous studies, and this structure can be further stabilized by the dynamic structural fluctuations due to the phason at finite temperatures. This suggests that the free energy of the $\Sigma 9$ GB can be extremely low at the high-temperature secondary

recrystallization process, and the $\Sigma 9$ GBs may become highly mobile even in the presence of secondary-phase pinning particles. Moreover, our DFT calculation predicted that Si atoms do not segregate to the stable $\Sigma 9 \{221\}$ GB structure (Supplementary Fig. 3). This tendency may also result in high mobility of the GB. However, to understand the actual GB migration process in detail, GB migration simulations or controlled experiments should be necessary, which will be our future plan.

Fig.R3: The segregation energies of Si at $\Sigma 9 \{221\}$ GB. **a**, The atomic structure of $\Sigma 9 \{221\}$ GB. The black triangle indicates the center of GB. The Fe atoms at the center and two different layers along the $[110]$ viewing direction are colored red, blue, and light blue, respectively. **b**, The segregation energies of substitutional Si in Fe sites as a function of distance from the center of the GB. The atoms in the same atomic columns are surrounded by blue rectangles. The red arrow corresponds to the most stable segregation site.

Response to the Reviewer #2

We thank the reviewer for reviewing our manuscript. We are very gratified that the reviewer basically agrees with us on the originality, novelty and significance of our findings. However, the reviewer raised several concerns and questions on our manuscript. We have carefully considered all the criticisms and comments raised by the referee, and revised our manuscript accordingly by incorporating new data in order to address all these concerns. The summary of our responses to the points raised by the reviewer and the corresponding changes made to the manuscript are italicised and interspersed between the reviewer's report below. In the revised manuscript, we highlight how we revised our manuscript by red colour font.

1. This is an interesting and stimulating paper in which state-of-the-art experimental high-resolution microscopy of two different σ_9 grain boundaries in bcc Fe are compared. The presence of 2% Si in the experimental samples is ignored for the purposes of analysis with theoretical models, which is a reasonable for a first study of this kind, which suggests that the free energy of phasons is an important stabilizing mechanism in the $\Sigma_9\{221\}$ grain boundary.

We thank the reviewer for this very positive comment.

2. The upper-case notation in Fig.3 used for labelling sub-figures, is inconsistent with the lower-case notation in the caption. The figure would be less confusing if subfigures were also labelled in lower case. Upper case letters I, D, P are also used in Fig.3 to label structures within sub-figures A, D, C, but not in sub-figures D, E, F. This should be made consistent within the figure and also consistent with the notation in the body of the text. The same inconsistent use of upper or lower case occurs in the Extended data figures, but it is less confusing there.

We thank the reviewer for this helpful comment. We have corrected figure labelling in Fig.3 and supporting figures, and have made consistency throughout the paper.

3. A particular problem with the paper is that there is insufficient consideration of the precision of the calculations, meaning the sources of error in grain boundary energies (and structures) apart from the well-known inaccuracy of empirical potentials or DFT. We are told that this is a slab calculation with periodic boundary conditions parallel to the grain boundary and free surfaces on the outside of the slab. Since the boundaries are symmetric the two free surfaces are the same. Presumably the atoms at the free surface and for some depth into the slab are not relaxed, or perhaps they are. In either case I can't find any discussion of the influence on the results of the thickness of the slab, and from experience I know that this should be much greater than the periodicity along the boundary. Is the system also periodically repeated perpendicularly to the grain boundary, and, if so, how thick is the vacuum region?

In the present calculation, we used the model with double grain boundaries (GBs) instead of slab model. In this model, there are two equivalent GBs in the supercell, and no vacuum layers are introduced. This point is not clear in the original manuscript, so we revised the manuscript as follows.

In the present calculation, we used the simulation cell with double grain boundaries instead of slab model. In this model, there are two equivalent GBs in the supercell, and no vacuum layers are introduced.

The paper includes an investigation of varying the supercell size along the [110] axis but not along an orthogonal direction parallel to the boundary, which should be no less worthy of testing.

In the model with double GBs, the distance between GBs can influence the estimated GB energy. We evaluated the GB energy of 1×3 GB as a function of distance between GBs. Fig.R4 shows the result. The simulation is done with ADP potential. The distance between the GBs was set to be about 17.00 \AA for all the $\Sigma 9 \{221\}$ calculations shown in this paper. When the distance is increased to the larger value of 42.00 \AA , the GB energy decreases by only 0.02 J m^{-2} . We think that this difference is negligibly small and thus the distance between the GBs of about 17.00 \AA is enough to evaluate GB structures and energies in the present study.

This figure is included as Supplementary Fig.S12.

Fig.R4: The relationship between the GB distance and the GB energy in $\Sigma 9 \{221\}$ (1×3) GB.

4. With reference to the relaxed structures obtained with the 1×3 , 1×5 and 1×8 repeat units after simulated annealing we read (line 117): “Further relaxations in the DFT calculations

did not show significant changes to the obtained structures”. This is curious, given that, according to Extended Data Table 1, although the energy of the 1 x 3 is lower than that of the 1 x 5 by only 5 mJ/m² when both are calculated with simulated annealing, the further relaxed DFT energy of the 1 x 3 is 101 mJ/m² lower than that of the 1 x 5. I believe 5 mJ/m² would be within the error bars of the calculation, and structures that appear to be separated by this energy could well be equivalent. However, 101mJ/m² stands out as indicating a significant difference in structure. For example, stacking fault energies are typically of this order of magnitude. This is a particular example for which more testing and reporting of the precision of calculations might be illuminating.

Since the GB atomic structure of 1×8 cell is the same as that combining the GB structures of 1×3 and 1×5 cells, it is simply expected that the GB energy of 1×8 cell should become smaller than that of 1×5 cell. However, the calculated GB energies are almost comparable in the DFT case. This discrepancy suggests that some of the DFT results shown here may be trapped in the metastable magnetic states. In DFT calculation, the magnetic state of the system is strongly influenced by the initial configuration of magnetic moments. In this study, we constructed the initial spin configuration assuming that all spins are aligned (ferromagnetic). However, if, for example, the spins of some Fe atoms in the GB core are more stable in antiferromagnetic-like state, the above assumption is not correct and may end up with metastable spin states with higher GB energies. Due to the calculation cost, we could not test all the possible initial configurations of spins at Fe atoms for DFT structural relaxation. Therefore, we cannot exclude the possibility that the obtained GB energies are trapped in the metastable spin configurations.

To overcome this problem, we newly performed MD simulations using recently developed Neural-Network-Potential (NNP) by Mori and Ozaki (Phys. Rev. Mater., 4 (2020), p. 40601). NNP has more flexible function form than ADP, and a well-tuned NNP achieves higher accuracy than conventional interatomic potentials such as ADP. Although we cannot directly specify the spin states of the system, the spin is indirectly incorporated into the potential by fitting to the results of DFT calculations with various spin states. Therefore, NNP can imitate DFT calculation including the effect of spin configurations and find the global energy minimum structures. The calculated results are shown in Fig.R5. It is found that the basic trend of energy vs. cell size and the obtained GB atomic structures are the same with the results of ADP. However, the absolute energy values become comparable to the DFT results, showing better energy accuracy. The present results also support that the incommensurate structure is energetically most stable structure for the present $\Sigma 9$ grain boundary.

We have included the Fig.R5 as Supplementary Fig.6 and pointed out the possible trapping at metastable magnetic structures in the DFT results.

Fig.R5: GB energies of the most stable GB structures for each cell size derived by ADP, DFT and NNP.

5. With reference to the phason free energy, there is no doubt that the phasons contribute something to the stabilisation of the structure at finite temperature, but they make no contribution to its stabilisation in the zero temperature DFT calculations presented here. Even at zero temperature, phonon zero-point energies, not considered here, with or without phasons present, make a difference to the free energy.

We thank the reviewer for this intriguing comment. Since we are interested in the finite temperature free energy affected by the phasons, we have newly calculated the entropy effect of the phason with 1×8 simulation cell. The results are explained in detail in the next question. Phonon zero-point energies at zero temperature may be needed to precisely evaluate the DFT calculated energies at zero temperature, but beyond our scope of the present study.

6. No evidence is given to support the statement “Thus, the free energy of the $\Sigma 9\{221\}$ GB can be much lower than other GBs at finite temperatures”; that is to say, there is no data presented to indicate that “much lower” makes the difference between a metastable structure and a most stable structure. The role of phasons needs to be more quantitatively identified in the model grain boundary energy, even though in real materials this would be experimentally very difficult. Note that anharmonicity and electronic excitations always cause the specific heat of a crystalline material to exceed the Dulong-Petit limit at high temperature, and quantisation of phonon energy cause it to be lower than this limit

at low temperature, with the outcome that the Dulong-Petit limit is mostly observed as little more than a point of inflection in the specific heat versus temperature of a perfect crystal. And the free energy of perfectly periodic grain boundaries generally tends to decrease with temperature without phasons. So the statement on line 178: “As for quasicrystals, specific heat exceeds the Dulong–Petit value at high temperatures due to the phason” is not helpful to the case, without some quantitative information and comparison with the other effects that are relevant at finite temperature.

We evaluated the GB free energies, $F_{GB}(T)$, using the non-equilibrium Frenkel-Ladd method (D. Frenkel et al., J. Chem. Phys. 81, 3188 (1984), M. Watanabe et al., Phys. Rev. Lett. 65, 3301 (1990), R. Freitas et al., Comput. Mater. Sci. 112, 333 (2016)), which is a kind of switching Hamiltonian method. We performed all the free energy calculations with ADP potential. Fig.R2 shows the GB free energies of $\Sigma 9 \{221\} (1 \times 8)$, $\Sigma 9 \{221\} (1 \times 1)$, and $\Sigma 9 \{114\}$ GBs as a function of temperature. The $\Sigma 9 \{221\} (1 \times 8)$ has much lower GB free energy in all the temperature range. The temperature dependences of the GB free energies are approximated with the straight lines. The GB entropies obtained from the slope of $F_{GB}(T)$ for $\Sigma 9 \{221\} (1 \times 8)$, $\Sigma 9 \{221\} (1 \times 1)$, and $\Sigma 9 \{114\}$ are 2.37×10^{-1} , 1.39×10^{-1} , and $8.45 \times 10^{-2} \text{ mJ m}^{-2} \text{ K}^{-1}$, respectively. The $\Sigma 9 \{221\} (1 \times 8)$ GB has a much higher GB entropy than other GBs. The present results clearly show that the present incommensurate GB structure has entropic stabilization effect at high temperatures. We included the Fig.R2 as the new Fig.4 and also included corresponding discussion in the main manuscript as follows.

To investigate the entropic contribution from the phason in the present GB, we evaluated the GB free energies, $F_{GB}(T)$, using the non-equilibrium Frenkel-Ladd method, which is a kind of switching Hamiltonian method. We performed all the free energy calculations with ADP potential. Fig.4 shows the GB free energies of $\Sigma 9 \{221\} (1 \times 8 \text{ cell})$, $\Sigma 9 \{221\} (1 \times 1 \text{ cell})$, and $\Sigma 9 \{114\}$ GBs as a function of temperature. The $\Sigma 9 \{221\} (1 \times 8 \text{ cell})$ has much lower GB free energy in all the temperature range. The temperature dependences of the GB free energies are approximated with the straight lines. The GB entropies obtained from the slope of $F_{GB}(T)$ for $\Sigma 9 \{221\} (1 \times 8 \text{ cell})$, $\Sigma 9 \{221\} (1 \times 1 \text{ cell})$, and $\Sigma 9 \{114\}$ are 2.37×10^{-1} , 1.39×10^{-1} , and $8.45 \times 10^{-2} \text{ mJ m}^{-2} \text{ K}^{-1}$, respectively. The $\Sigma 9 \{221\} (1 \times 8 \text{ cell})$ GB has a much higher GB entropy than the other GBs. The present results clearly show that the incommensurate GB structure has entropic stabilization effect at finite temperatures.

Fig.R2: GB free energies as a function of temperature. The error bars correspond to the standard deviations of 10 independent simulations.

7. A minor typo in lines 115 and 125: alternative -> alternating

We have corrected the wording in the revised manuscript.

Response to the Reviewer #3

We thank the reviewer for reviewing our manuscript. The reviewer raised several concerns on our manuscript. We have carefully considered all the criticisms and comments raised by the referee, and revised our manuscript accordingly by incorporating new data, in order to address all these concerns. The summary of our responses to the points raised by the reviewer and the corresponding changes made to the manuscript are italicised and interspersed between the reviewer's report below. In the revised manuscript, we highlight how we revised our manuscript by red colour font.

The authors reported on an incommensurate grain-boundary atomic structure of $\Sigma_9\{221\}$ STGB in Fe-Si alloy based on STEM and atomistic simulations. They also proposed an unconventional entropic stabilization mechanism at finite temperature due to an additional lattice excitation (i.e. phason). However, I disagree with most of their arguments.

1. the authors said there is no Si segregation in the GB (based on EELS). Thus the reported structure should be independent on Si segregation. This is hardly believed.

Based on this criticism, we have performed DFT calculation of Si segregation energies in this GB. The results are shown in Fig.R3. Here, we evaluated the Si segregation energies for all the Fe atomic sites in the core region (labeled A to F). Our results show that the segregation energies of Si become positive for all the GB sites, suggesting that Si atoms do not segregate to the present GB. These results are in agreement with our EELS experiment. We included the Fig.R3 as the Supplementary Fig.3.

Fig.R3: The segregation energies of Si at $\Sigma_9\{221\}$ GB. **a**, The atomic structure of $\Sigma_9\{221\}$ GB. The black triangle indicates the center of GB. The Fe atoms at the center and two different layers along the $[110]$ viewing direction are colored red, blue, and light blue, respectively. **b**, The segregation energies of substitutional Si in Fe sites as a function of distance from the center of the GB. The atoms in the same atomic columns are surrounded by blue rectangles. The red arrow corresponds to the most stable segregation site.

2. since they don't consider Si role in GBs, all simulations were conducted for PURE Fe. The authors can examine Si solubility in GB compared with in grains. If Si is preferred to segregated into GBs, then the author must prove Si segregation does not influence the B structure.

Our DFT results show that Si atoms tend not to segregate to any of the GB sites in the present GB. We think this is one of the reasons why our pure Fe based calculation show excellent agreement with the experimental observation.

3. Since there is no reliable empirical potential for Al-Si system, all simulations should be done by using DFT. The authors should consider the size effect on DFT calculations, not only the unit cells, but also the size of the two grains because of either two free surfaces or double GBs in the model.

We agree that there is no good empirical potential for Fe-Si system. Since we have shown that Si atoms do not segregate to the GB by DFT and EELS, we use pure Fe potential and MD for the present theoretical calculations. This allows us to explore the stable GB structures with larger cell sizes. Doing all the simulations with DFT may be desirable, but computationally this is not feasible because large simulation cells are needed for reproducing the present GB structure. Moreover, as noted in the response letter for the reviewer #2, DFT results can be trapped in the metastable Fe spin configurations if we cannot test vast initial spin configuration for structural relaxation.

In our previous submission, we used angular dependent potential (ADP) of pure Fe. It is pointed out that ADP potential sometimes underestimates GB energies compared with DFT. Therefore, we redone our MD simulation using recently-reported Neural-Network Potential (NNP) by Mori and Ozaki (Phys. Rev. Mater., 4 (2020), p. 40601) to estimate the stable GB structures and energies. NNP has more flexible function form than ADP, and a well-tuned NNP achieves higher accuracy than conventional interatomic potentials such as ADP. Although we cannot directly specify the spin states of the system, the spin is indirectly incorporated into the potential by fitting to the results of DFT calculations with various spin states. Therefore, NNP can imitate DFT calculation including the effect of spin configurations and find the global energy minimum structures. The calculated results are shown in Fig.R5. It is found that the basic trend of energy vs. cell size and the obtained GB atomic structures are the same with the results of ADP. However, the absolute energy values become comparable to the DFT results, showing better energy accuracy. The present results also support that the incommensurate structure is energetically most stable structure for the present $\Sigma 9$ grain boundary. We have included the Fig.R5 as Supplementary Fig.6 and pointed out the possible trapping at metastable magnetic structures in the DFT results. Since the stable GB atomic structures are basically the same in the both potential models (ADP and NNP), ADP results are kept used in the main text.

Fig.R5: GB energies of the most stable GB structures for each cell size derived by ADP, DFT and NNP.

Response to the Reviewer #4

We thank the reviewer for reviewing our manuscript. We are very gratified that the reviewer basically agrees with us on the significance of our findings. However, the reviewer raised several concerns and questions on our manuscript. We have carefully considered all the criticisms and comments raised by the referee, and revised our manuscript accordingly by incorporating new data in order to address all these concerns. The summary of our responses to the points raised by the reviewer and the corresponding changes made to the manuscript are italicised and interspersed between the reviewer's report below. In the revised manuscript, we highlight how we revised our manuscript by red colour font.

I like your expertized work with target-oriented preparation of bicrystals, cutting edge STEM and detailed theoretical calculations to interpret the observations and consequences of them.

Your observation that proper understanding of the symmetric Sigma9/221 in bcc Fe, meaning finding its stable lowest energy state, needs multiple lattice periodic units along the tilt axis is remarkable. This justifies publication as it clearly demonstrates that the usual practice of simulating GB structures in rather small depth periodic units can be misleading for unfortunate choice of grain boundary. On the other hand, it might not be too suprising, considering the fact that all known investigations naturally use several lattice units perpendicular to the axis when constructing a supercel. To the bottom end, a GB is a two dimensional surface, for which it is likely that the supercell has to be extended in both lateral dimension to resonably approximate the structure.

We thank the reviewer for this positive comment. This comment clearly highlights the main finding of the present study.

It is a very clear result that your newly identified structure, described as a sequence of ikosahedral, double ikoshedral clusters and biprisyms, has a lower energy, convincingly matches to your STEM images, and your quantitative values on the GB excess energy appear reasonable. I can well understand that the equilibrium structure could be only found with the help of simulated annealing, while conventional DFT search missed it. But after it has been found, I do not understand that the corresponding DFT energies (third column of table in extended material) do not show the energy reduction with reference to 1×1 . If there is no simple error in the presented numbers, the origin of this calculation must be better explained in the article to clarify the situation.

Since the GB atomic structure of 1×8 cell is the same as that combining the GB structures of 1×3 and 1×5 cells, it is simply expected that the GB energy of 1×8 cell should become much smaller than that of 1×5 cell. However, the calculated GB energies are almost comparable in the present DFT case. This discrepancy suggests that some of the DFT results shown here may be trapped in the metastable magnetic states. In DFT calculation, the magnetic state of the system is strongly influenced by the initial configuration of

magnetic moments. In this study, we constructed the initial spin configuration assuming that all spins are aligned (ferromagnetic). However, if, for example, the spins of some Fe atoms in the GB core are more stable in the antiferromagnetic-like state, the above assumption is not correct and may end up with metastable spin states with higher GB energies. Due to the calculation cost, we could not test all the possible initial configurations of spins at Fe atoms for DFT structural relaxation. Therefore, we cannot exclude the possibility that the obtained GB energies are trapped in the metastable spin configurations in the present DFT results.

To overcome this problem, we newly performed MD simulations using recently developed Neural-Network-Potential (NNP) by Mori and Ozaki (*Phys. Rev. Mater.*, 4 (2020), p. 40601). NNP has more flexible function form than ADP, and is a well-tuned NNP achieves higher accuracy than conventional interatomic potentials such as ADP. Although we cannot directly specify the spin states of the system, the spin is indirectly incorporated into the potential by fitting to the results of DFT calculations with various spin states. Therefore, NNP can imitate DFT calculation including the effect of spin configurations and find the global energy minimum structures. The calculated results are shown in Fig.R5. It is found that the basic trend of energy vs. cell size and the obtained GB atomic structures are the same with the results of ADP. However, the absolute energy values become comparable to the DFT results, showing better energy accuracy.

In the Supplementary Fig.6, we explain the possible origin of why the DFT energies do not show the energy reduction with reference to 1x1 and show the new results with NNP.

Fig.R5: GB energies of the most stable GB structures for each cell size derived by ADP, DFT and NNP.

By contrast, I am less confident that the shown evidence for the phasons and the consequent stabilization of Gibbs energy by additional entropic contributions, is sufficient

to be convincing. Yes, it is clear that the low energy configuration of the GB needs a longer sequence of different building blocks. Also the total length of this sequence might be incommensurate to the initial lattice periodicity, although I miss a real proof for this in the article. But most important, the three snapshots in Fig. 4 just essentially show three times the same structure, if one takes into account the periodic boundary conditions between top and bottom of the simulated volume. So what you interpret as a phason could simply be a numerical effect giving a slight shift to the position with respect to the position of the periodic boundaries.

In order to answer these very important comments, we evaluated the GB free energies, $F_{GB}(T)$, using the non-equilibrium Frenkel-Ladd method (D. Frenkel et al., J. Chem. Phys. 81, 3188 (1984), M. Watanabe et al., Phys. Rev. Lett. 65, 3301 (1990), R. Freitas et al., Comput. Mater. Sci. 112, 333 (2016)), which is a kind of switching Hamiltonian method. We performed all the free energy calculations with ADP potential. Fig.R2 shows the GB free energies of $\Sigma9 \{221\} (1 \times 8)$, $\Sigma9 \{221\} (1 \times 1)$, and $\Sigma9\{114\}$ GBs as a function of temperature. The $\Sigma9 \{221\} (1 \times 8)$ has much lower GB free energy in all the temperature range. The temperature dependences of the GB free energies are approximated with the straight lines. The GB entropies obtained from the slope of $F_{GB}(T)$ for $\Sigma9 \{221\} (1 \times 8)$, $\Sigma9 \{221\} (1 \times 1)$, and $\Sigma9\{114\}$ are 2.37×10^{-1} , 1.39×10^{-1} , and $8.45 \times 10^{-2} \text{ mJ m}^{-2} \text{ K}^{-1}$, respectively. The $\Sigma9 \{221\} (1 \times 8)$ GB has a much higher GB entropy than other GBs. The present results clearly show that the incommensurate GB structure has entropic stabilization effect at high temperatures. We included the Fig.R2 as the new Fig.4 and also included corresponding discussion in the main manuscript as follows.

To investigate the entropic contribution from the phason in the present GB, we evaluated the GB free energies, $F_{GB}(T)$, using the non-equilibrium Frenkel-Ladd method, which is a kind of switching Hamiltonian method. We performed all the free energy calculations with ADP potential. Fig.4 shows the GB free energies of $\Sigma9 \{221\} (1 \times 8 \text{ cell})$, $\Sigma9 \{221\} (1 \times 1 \text{ cell})$, and $\Sigma9\{114\}$ GBs as a function of temperature. The $\Sigma9 \{221\} (1 \times 8 \text{ cell})$ has much lower GB free energy in all the temperature range. The temperature dependences of the GB free energies are approximated with the straight lines. The GB entropies obtained from the slope of $F_{GB}(T)$ for $\Sigma9 \{221\} (1 \times 8 \text{ cell})$, $\Sigma9 \{221\} (1 \times 1 \text{ cell})$, and $\Sigma9\{114\}$ are 2.37×10^{-1} , 1.39×10^{-1} , and $8.45 \times 10^{-2} \text{ mJ m}^{-2} \text{ K}^{-1}$, respectively. The $\Sigma9 \{221\} (1 \times 8 \text{ cell})$ GB has a much higher GB entropy than the other GBs. The present results clearly show that the incommensurate GB structure has entropic stabilization effect at finite temperatures.

Fig.R2: GB free energies as a function of temperature. The error bars correspond to the standard deviations of 10 independent simulations.

Secondly, we have performed an additional simulation similar to Fig. 4, but with a much larger simulation cell of 1×32 . The stable structure of 1×32 GB shown in Fig. R6a is not the repetition of shorter-period structures. The GB energy of 1×32 GB (1.357 J m^{-2}) is close to that of 1×8 GB (1.378 J m^{-2}), confirming that a very long or non-periodic structure can be formed at this GB. To further verify this point, we performed the MD simulation at 200 K. As shown in Fig.R6b and c, the core Fe atoms frequently moved during the simulation, and the GB atomic structures fluctuated between inequivalent structures from moment to moment. This is more clearly seen in Fig.R7, where time evolution of the core atom positions is tracked.

These results are included as Supplementary Fig.7 and 10.

*Fig.R6: The atomic structure of the long-period 1×32 cell GB. **a**, The stable structure of the 1×32 cell GB simulated with ADP potential. The core atoms are shown with red color and the labels *i* and *p* indicates the interlayer site and the center of pentagons, respectively. **b** and **c** are snapshots from constant temperature MD simulation at 14.50 ps and 15.00 ps, respectively. The red arrow indicates the sites at which large atomic fluctuations are observed.*

Fig.R7: Time evolution of the core atom positions in 1×32 cell GB at 200 K. From the trajectory of MD simulation, the core atom positions are classified as p (pentagonal) or i (interlayer), which are shown as red and blue filled circles, respectively. The core atoms are classified as p if their projected positions are within 0.3 \AA of the bulk projected positions. As the time evolves, some of the core atoms transit to different positions, making the overall GB atomic structure different before and after the transition.

If you would like to keep your statement of entropic stabilization and phasons, I feel that a controlled variation of the height of the simulated volume over a larger range, and quantitative comparison of the outcome with the Equation (referenced from 30) is necessary. Also a clear estimation/calculation of the amount of the entropic effect to the Gibbs energy would be required. As far as I understood, the presently applied theoretical methods are only suitable to calculate the energy/enthalpy. For calculation of the entropy or the Gibbs energy, either quasiharmonic approximation, switching hamiltonian methods (see e.g. doi:10.1016/j.actamat.2018.01.006) or similar should be applied to derive a clear prediction of the excess entropy of the GB.

We thank the reviewer for these very important suggestions and recommendation of the references. As already shown in the previous answer, we have simulated GB free energy including entropic contribution based on the suggested methods.

REVIEWER COMMENTS

Reviewer #1 (Remarks to the Author):

The authors have updated their manuscript and made several changes to clarify the comments of the reviewers. However, I still see open points on the explanation of the phases and the DFT calculations:

A) Author statement in Fig caption R1: “Fig.R1: The stable GB atomic structures of 1×2, 1×4, and 1×7 cells. a, b show the atomic structures of 1×2 (1×4) and 1×7 GBs observed along the [110] axis. Note that the 1×2 and 1×4 GBs have the same atomic structure ...”

Reviewer reply and concern: The authors show in Fig R1 that the “ 1×2 and 1×4 GBs have the same atomic structure”. This indicates in my opinion, that the argument of an incommensurate structure is not holding as the authors show that larger cells are similar to combinations of the smaller cells.

B) Author statement: We evaluated the GB free energies, $F_{GB}(T)$, using the non-equilibrium Frenkel-Ladd method (D. Frenkel et al., J. Chem. Phys. 81, 3188 (1984)., M. Watanabe et al., Phys. Rev. Lett. 65, 3301 (1990)., R. Freitas et al., Comput. Mater. Sci. 112, 333 (2016)), which is a kind of switching Hamiltonian method. We performed all the free energy calculations with ADP potential. Fig.R2 shows the GB free energies of $\Sigma 9 \{221\} (1 \times 8)$, $\Sigma 9 \{221\} (1 \times 1)$, and $\Sigma 9 \{114\}$ GBs as a function of temperature. The $\Sigma 9 \{221\} (1 \times 8)$ has much lower GB free energy in all the temperature range. The temperature dependences of the GB free energies are approximated with the straight lines. The GB entropies obtained from the slope of $F_{GB}(T)$ for $\Sigma 9 \{221\} (1 \times 8)$, $\Sigma 9 \{221\} (1 \times 1)$, and $\Sigma 9 \{114\}$ are 2.37×10^{-1} , 1.39×10^{-1} , and 8.45×10^{-2} mJ m⁻² K⁻¹, respectively. The $\Sigma 9 \{221\} (1 \times 8)$ GB has a much higher GB entropy than other GBs. The present results clearly show that the incommensurate GB structure has entropic stabilization effect at high temperatures. We include the Fig.R2 as the new Fig.4 and also included corresponding discussion in the main manuscript as follows.

Reviewer reply and concerns: The authors show that the (excess) free energy of the 1x1 reconstruction follows a different path than the 1x8 reconstruction (Fig R2). Later, one can see that the 0 K energy is not converged here, but for e.g. 1x3 or 1x5. A fair comparison would be with those, not with 1x1. Indeed, one would want to show a graph where the 0 K

energies are the same, but the slope would be steeper for a larger system (or something like this).

Additionally, non-equilibrium thermodynamic integration along the Frenkel–Ladd path works as follows: One starts from an Einstein crystal (independent harmonic oscillators fixed in space, no interaction), for which one has an analytical solution for the absolute free energy. Then we slowly switch the system to the more realistic description (EAM, DFT, whatever), while recording the work done on the system. Then we switch back to the Einstein crystal while again recording the work done on the system. The assumption is that there is an error due to dissipation, but that it is the same forward and backward and therefore cancels. Here, things (i.e. Fe atoms) start moving around in the realistic system, but the atoms are pinned in space in the Einstein crystal. This can lead to different dissipation errors in forward and backward direction. The authors should try to exclude this or show some data that verifies that their thermodynamic integration still works. Even if all the explanations of the authors are correct, can it be excluded that a local, new GB phonon mode causes the additional entropy instead of a phason? I would suggest to consider this possibility as well.

c) Author statement: “Since the GB atomic structure of 1×8 cell is the same as that combining the GB structures of 1×3 and 1×5 cells, it is simply expected that the GB energy of 1×8 cell should become smaller than that of 1×5 cell. However, the calculated GB energies are almost comparable in the DFT case. This discrepancy suggests that some of the DFT results shown here may be trapped in the metastable magnetic states.”

Reviewer reply: An alternative interpretation could be that the DFT energy truly converges because the reconstruction is big enough. Can you exclude this?

Reviewer #2 (Remarks to the Author):

This manuscript reports some beautiful experimental images and images simulated from MD calculations for the grain boundary they are studying. They really should be published. The revised manuscript makes it much clearer how they authors have proceeded in their their theoretical approach to understanding the energetics of these boundaries, which

present multiple arrangements of three basic structural units. The structural units can morph into each other by small but identifiable atomic displacements, with naturally low energy barriers between the resulting possible arrangements. These structures would thus be expected to exhibit very anharmonic lattice dynamics, so examining free energy differences with an efficient lambda-integration method makes good sense. However, the data still doesn't seem to warrant the emphasis on "phasons", since there is no attempt to estimate what the free energy of phasons might contribute to free energy above and beyond the conventional quasi-harmonic and anharmonic lattice models.

They are leaning heavily on Fig. 4, comparing the free energy versus temperature of the grain boundaries for which the supercell is constrained to just one period repeat, the blue line (1x1), with the larger supercell, (1x8) in which the multiple arrangements of structural units can be observed. It is not clear why the former free energy tends to a different value than the latter as $T \rightarrow 0$. They should tend to the corresponding internal energies, which are reported in the Supplementary material to be very similar, about 1.38. However, on Fig. 4. in the manuscript they differ by about 0.14, as I estimate from the graph.

Reviewer #3 (Remarks to the Author):

The authors addressed my concerns.

Reviewer #4 (Remarks to the Author):

My main criticism on the initial manuscript was the highly speculative argument regarding the entropic stabilization due to kinetic phason disorder. Obviously two other reviewers had also raised the same concern.

You now present detailed calculation of the Excess Gibbs energy and the excess entropy showing clearly that the GB with the larger unit cell has indeed a significantly higher excess entropy, which convincingly supports the idea of entropic stabilization. I found this a remarkable and interesting result. The manuscript has thus substantially improved. Also the two other criticized aspects, the more speculative discussion regarding the effect on texture evolution and remaining accuracy concerns regarding the simulation are no more

substantial, since the main result on the entropy is robust against different theoretical models (DFT and two different interaction potentials) and a speculative suggestion at the end of the article, clearly indicated as such, is exactly what drives discussion among scientists.

If I can give a final hint on precise wording: Strictly speaking the (1x8) GB is not stabilized against the other options by entropy, since already at $T=0\text{K}$, its energy ($=$ Gibbs energy) is way less than for the other two examples. Thus, the entropy obviously gives only an additional contribution to the high stability.

Response to the Reviewer #1

We thank the reviewer for reviewing our manuscript again. The reviewer still raised several concerns and questions on our manuscript. We have carefully considered all the criticisms and comments raised by the referee, and revised our manuscript accordingly by incorporating new data in order to address all these concerns. The summary of our responses to the points raised by the reviewer and the corresponding changes made to the manuscript are italicised and interspersed between the reviewer's report below. In the revised manuscript, we highlight how we revised our manuscript by red colour font.

The authors have updated their manuscript and made several changes to clarify the comments of the reviewers. However, I still see open points on the explanation of the phasons and the DFT calculations:

A) Author statement in Fig caption R1: “Fig.R1: The stable GB atomic structures of 1×2 , 1×4 , and 1×7 cells. a, b show the atomic structures of 1×2 (1×4) and 1×7 GBs observed along the $[110]$ axis. Note that the 1×2 and 1×4 GBs have the same atomic structure ...”

Reviewer reply and concern: The authors show in Fig R1 that the “ 1×2 and 1×4 GBs have the same atomic structure”. This indicates in my opinion, that the argument of an incommensurate structure is not holding as the authors show that larger cells are similar to combinations of the smaller cells.

In the case of 1×2 and 1×4 GB cell sizes, the cell sizes are not large enough to exhibit incommensurate structure (and their GB energies are higher than that of 1×8 cell GB structure as shown in Fig.2). However, in the 1×8 cell size or larger (1×32 cell size), although the building blocks of the clusters are similar, their arrangement along the GBs shows long-range periodicity. Indeed, the 1×8 structure is not the repeat structures of 1×2 or 1×4 cell structures anymore. In the real GBs, there are no periodic boundary conditions imposed along the GBs which are required for simulating GB structures in computers. Therefore, we believe that, if the cell sizes become larger and larger, the GB structures exhibit very long-range periodicity and can be considered as incommensurate structure in the real GB.

B) Author statement: We evaluated the GB free energies, $F_{GB}(T)$, using the non-equilibrium Frenkel-Ladd method (D. Frenkel et al., J. Chem. Phys. 81, 3188 (1984)., M. Watanabe et al., Phys. Rev. Lett. 65, 3301 (1990)., R. Freitas et al., Comput. Mater. Sci. 112, 333 (2016)), which is a kind of switching Hamiltonian method. We performed all the free energy calculations with ADP potential. Fig.R2 shows the GB free energies of $\Sigma 9 \{221\}$ (1×8), $\Sigma 9 \{221\}$ (1×1), and $\Sigma 9 \{114\}$ GBs as a function of temperature. The $\Sigma 9 \{221\}$ (1×8) has much lower GB free energy in all the temperature range. The temperature

dependences of the GB free energies are approximated with the straight lines. The GB entropies obtained from the slope of $F_{GB}(T)$ for $\Sigma9 \{221\} (1 \times 8)$, $\Sigma9 \{221\} (1 \times 1)$, and $\Sigma9 \{114\}$ are 2.37×10^{-1} , 1.39×10^{-1} , and $8.45 \times 10^{-2} \text{ mJ m}^{-2} \text{ K}^{-1}$, respectively. The $\Sigma9 \{221\} (1 \times 8)$ GB has a much higher GB entropy than other GBs. The present results clearly show that the incommensurate GB structure has entropic stabilization effect at high temperatures. We include the Fig.R2 as the new Fig.4 and also included corresponding discussion in the main manuscript as follows.

Reviewer reply and concerns: The authors show that the (excess) free energy of the 1×1 reconstruction follows a different path than the 1×8 reconstruction (Fig R2). Later, one can see that the 0 K energy is not converged here, but for e.g. 1×3 or 1×5 . A fair comparison would be with those, not with 1×1 . Indeed, one would want to show a graph where the 0 K energies are the same, but the slope would be steeper for a larger system (or something like this).

Additionally, non-equilibrium thermodynamic integration along the Frenkel–Ladd path works as follows: One starts from an Einstein crystal (independent harmonic oscillators fixed in space, no interaction), for which one has an analytical solution for the absolute free energy. Then we slowly switch the system to the more realistic description (EAM, DFT, whatever), while recording the work done on the system. Then we switch back to the Einstein crystal while again recording the work done on the system. The assumption is that there is an error due to dissipation, but that it is the same forward and backward and therefore cancels. Here, things (i.e. Fe atoms) start moving around in the realistic system, but the atoms are pinned in space in the Einstein crystal. This can lead to different dissipation errors in forward and backward direction. The authors should try to exclude this or show some data that verifies that their thermodynamic integration still works.

Even if all the explanations of the authors are correct, can it be excluded that a local, new GB phonon mode causes the additional entropy instead of a phason? I would suggest to consider this possibility as well.

To fully address the above criticisms, we think further extensive theoretical analysis should be necessary. Since our main result of this study is the new finding of incommensurate GB atomic structure by both experiment and theory, we decided to remove Fig.4 and the discussion on the entropic stabilization effects by phason from the main manuscript, also following the suggestion from the editor. In our future studies, we would like to unambiguously confirm the entropic effects for the present GB structures by considering the above reviewer's suggestions and comments. We thank the reviewer for these constructive comments.

c) Author statement: “Since the GB atomic structure of 1×8 cell is the same as that

combining the GB structures of 1×3 and 1×5 cells, it is simply expected that the GB energy of 1×8 cell should become smaller than that of 1×5 cell. However, the calculated GB energies are almost comparable in the DFT case. This discrepancy suggests that some of the DFT results shown here may be trapped in the metastable magnetic states.”

Reviewer reply: An alternative interpretation could be that the DFT energy truly converges because the reconstruction is big enough. Can you exclude this?

First, we thoroughly checked the structural relaxation processes in DFT calculations again. We found that the 1×3 structure shows a cell shape transformation during the relaxation, which should have caused the energy error in our previous data. In other structures, we confirmed that the cell shape transformation did not occur. Therefore, in the revised Supplementary Fig.6, we corrected the GB energy of the 1×3 structure obtained by DFT calculation. The corrected GB energy of the 1×3 structure becomes higher than that of the previous calculation. However, the GB energies of the 1×3 structure and other larger cell structures are still lower than that of the 1×1 structure.

*Supplementary Fig. 6: GB energies of the most stable GB structures for each cell size derived by MD (ADP, NNP) and DFT. It is found that the basic trend of energy vs. cell size and the obtained GB atomic structures are the same in the results of ADP and NNP. However, DFT results show some discrepancy. **For DFT calculations, we assumed ferromagnetic spin configuration as the initial structures before structural relaxation. Here, we cannot exclude the possibility that the obtained GB energies are trapped in the metastable spin configurations by the present DFT calculations, as demonstrated in Supplementary Fig.13.** On the other hand, NNP has more flexible function form than ADP, and a well-tuned NNP achieves higher accuracy than conventional interatomic potentials such as ADP. Although we cannot directly specify the spin states of the system,*

the spin is indirectly incorporated into the potential by fitting to the results of DFT calculations with various spin states. Therefore, NNP can imitate DFT calculation including the effect of spin configurations and find the global energy minimum structures. The absolute energy values become comparable to the DFT results, showing better energy accuracy.

We still believe that the initial magnetic moments and their configuration should strongly influence the final magnetic structure and the GB energy. Therefore, our obtained GB energies by DFT shown in Supplementary Fig.6 should be trapped to the metastable ones. To demonstrate this possibility, we performed additional DFT calculations as follows. In the DFT results shown in Supplementary Fig.6, we assumed the ferromagnetic spin configuration and set the initial magnetic moments of all the Fe atoms to be 3.0. In the additional calculation, we changed the initial magnetic moments of the Fe atoms in the GB core sites to be 0.0 or -3.0, while the magnetic moments of other sites are kept to be 3.0. Supplementary Fig.13 shows the relationship between the initial magnetic configurations and the relaxed GB energies for **b** 1×3 , **c** 1×5 , and **d** 1×8 GBs. The blue and orange lines correspond to the initial magnetic moments of 0.0 and -3.0, respectively. It is clear that the GB energies vary dramatically depending on the initial magnetic moments and configurations. In some magnetic configurations, the GB energies become lower than those assuming ferromagnetic spin configuration shown by the dashed lines. It should be noted that the relaxed GB atomic structures are almost the same regardless of the initial magnetic moments and configurations. Therefore, in order to find the global energy minimum magnetic structures by DFT, we need to consider all the possible initial magnetic moments and configurations in the GB structures, which is very interesting but beyond the scope of our present study. We included the Supplementary Fig.13 in the revised Supplementary information.

Supplementary Fig. 13: The GB energies calculated from the different initial magnetic moments and configurations by DFT. The initial magnetic moments of the Fe atoms as indicated by sites from 1 to 6 in **a** are gradually changed to be 0.0 or -3.0, while keeping the other atoms' magnetic moment to be 3.0. **b**, **c**, and **d** show the GB energies of 1×3 , 1×5 , and 1×8 GB, respectively, calculated from the different initial magnetic moments and configurations. The blue and orange lines correspond to the initial magnetic moments of 0.0 and -3.0, respectively. The dashed lines show the GB energy calculated

from the ferromagnetic magnetic configurations as shown in Supplementary Fig.6, where the initial magnetic moments are set to be 3.0 for all the Fe atoms.

Response to the Reviewer #2

We thank the reviewer for reviewing our manuscript again. We are very gratified that the reviewer is very positive for the publication of our results. We have carefully considered all the criticisms and comments raised by the referee, and revised our manuscript accordingly. The summary of our responses to the points raised by the reviewer are italicised and interspersed between the reviewer's report below. In the revised manuscript, we highlight how we revised our manuscript by red colour font.

This manuscript reports some beautiful experimental images and images simulated from MD calculations for the grain boundary they are studying. They really should be published.

We thank the reviewer for this very strong support for our results.

The revised manuscript makes it much clearer how they authors have proceeded in their theoretical approach to understanding the energetics of these boundaries, which present multiple arrangements of three basic structural units. The structural units can morph into each other by small but identifiable atomic displacements, with naturally low energy barriers between the resulting possible arrangements. These structures would thus be expected to exhibit very anharmonic lattice dynamics, so examining free energy differences with an efficient lambda-integration method makes good sense. However, the data still doesn't seem to warrant the emphasis on "phasons", since there is no attempt to estimate what the free energy of phasons might contribute to free energy above and beyond the conventional quasi-harmonic and anharmonic lattice models.

According this criticism and also the reviewer #1's criticisms, we admit that our discussion on the entropic stabilization by phason is not indisputable enough at the present stage, and further extensive theoretical analysis should be needed to unambiguously confirm the entropic effects by phason. Since our main result of this study is the new finding of incommensurate GB atomic structure, we decided to remove Fig.4 and the discussion on the entropic stabilization effects by phason from the main manuscript, also following the suggestion from the editor. In our future studies, we would like to unambiguously confirm the entropic effects by phason through answering the above reviewer's comments. Thank you for these comments.

They are leaning heavily on Fig. 4, comparing the free energy versus temperature of the grain boundaries for which the supercell is constrained to just one period repeat, the blue line (1x1), with the larger supercell, (1x8) in which the multiple arrangements of structural units can be observed. It is not clear why the former free energy tends to a different value than the latter as $T \rightarrow 0$. They should tend to the corresponding internal

energies, which are reported in the Supplementary material to be very similar, about 1.38. However, on Fig. 4. in the manuscript they differ by about 0.14, as I estimate from the graph.

The GB energies at 0K for the (1×1) and (1×8) structures are estimated to be 1.58 and 1.38 J/m², respectively, using ADP potential as shown in the Supplementary Table 1. This difference is consistent with the difference between the GB energies of (1×1) and (1×8) structures at 0K as shown in Fig.4.

Although we think our results shown in Fig.4 are consistent, we decided to remove Fig.4 and the discussion on phason in the revised manuscript as discussed above.

Response to the Reviewer #4

We thank the reviewer for reviewing our manuscript again. We are very gratified that the reviewer agrees with us on the significance of our findings. In the following, we respond to the minor comment raised by the reviewer.

My main criticism on the initial manuscript was the highly speculative argument regarding the entropic stabilization due to kinetic phason disorder. Obviously two other reviewers had also raised the same concern.

You now present detailed calculation of the Excess Gibbs energy and the excess entropy showing clearly that the GB with the larger unit cell has indeed a significantly higher excess entropy, which convincingly supports the idea of entropic stabilization. I found this a remarkable and interesting result. The manuscript has thus substantially improved. Also the two other criticized aspects, the more speculative discussion regarding the effect on texture evolution and remaining accuracy concerns regarding the simulation are no more substantial, since the main result on the entropy is robust against different theoretical models (DFT and two different interaction potentials) and a speculative suggestion at the end of the article, clearly indicated as such, is exactly what drives discussion among scientists.

We thank the reviewer for highly evaluating our previous revisions.

If I can give a final hint on precise wording: Strictly speaking the (1x8) GB is not stabilized against the other options by entropy, since already at $T=0\text{K}$, its energy (=Gibbs energy) is way less than for the other two examples. Thus, the entropy obviously gives only an additional contribution to the high stability.

We thank the reviewer for this useful suggestion. According to the further criticisms by the reviewer #1 and #2, we admit that further extensive theoretical analysis should be needed to unambiguously confirm the entropic effects by phason. Since our main result of this study is the new finding of incommensurate GB atomic structure, we decided to remove Fig.4 and the discussion on the entropic stabilization effects by phason from the manuscript, also following the suggestion by the editor. In our future studies, we would like to unambiguously confirm the entropic effects by phason with further theoretical analysis.

REVIEWERS' COMMENTS

Reviewer #1 (Remarks to the Author):

The paper reports interesting and novel results of an incommensurate GB structure in Fe-Si.

The revised version is now suitable for publication in Nature Comm.